# Federated Multi-view Clustering for Remote Sensing Data

**Renxiang Guan** [1]  **Xiang Yang** [2]  **Hao Yu** [1]  **Siwei Wang** [3]  **Suyuan Liu** [1]  **Wenjing Yang** [1]  **Junjie Huang** [1]  **Ao Li** [4]
**Xinwang Liu** [1]  **Yuhua Tang** [1]

## Abstract

The rapid expansion of remote sensing technology has generated massive amounts of unlabeled multi-view data distributed across different institutions. Analyzing this data presents significant challenges, as centralized processing incurs prohibitive communication costs and raises data privacy concerns. To address these issues, this paper proposes a novel deep federated multi-view clustering (MVC) framework tailored for remote sensing data. Unlike existing methods that transmit sensitive data features, our approach shares only privatized prototypes masked with adaptive noise, ensuring both communication efficiency and privacy preservation. First, we employ superpixel segmentation to reduce the spatial dimensionality of remote sensing data, lowering computational burdens. Furthermore, to resolve the inconsistency of cluster assignments across different clients, we design a co-occurrence structural alignment module that synchronizes local models. Finally, we incorporate a wasserstein prototype contrastive learning mechanism, which models clusters as distributions rather than points, to enhance global consistency and robustness against data heterogeneity. Extensive experiments on four public datasets demonstrate that our framework achieves superior clustering performance and efficiency compared to state-of-the-art methods. The source code is available here: https://github.com/GuanRX/FedRSMVC.

[1] College of Computer Science and Technology, National University of Defense Technology, Changsha, China [2] School of Computer Science and Technology, South-Central Minzu University, Wuhan, China [3] Academy of Military Sciences, Beijing, China [4] School of Computer Science and Technology, Harbin University of Science and Technology, Harbin, China. Correspondence to: Wenjing Yang <wenjing.yang@nudt.edu.cn; renxiangguan@nudt.edu.cn>.

*Proceedings of the 43rd International Conference on Machine Learning*, Seoul, South Korea. PMLR 306, 2026. Copyright 2026 by the author(s).

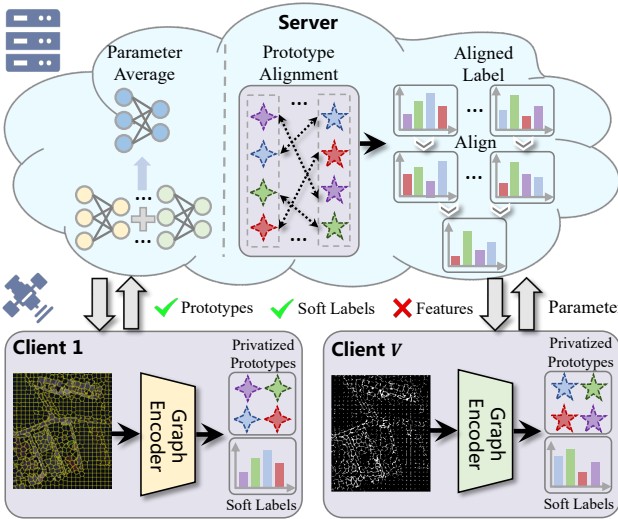

*Figure 1.* Schematic diagram of the proposed deep FMVC framework. Unlike existing methods that transmit sensitive features (marked with a cross), our approach shares only privatized prototypes masked with adaptive noise and soft labels (marked with checks). This design ensures privacy preservation and high communication efficiency by avoiding raw data transmission.

## 1. Introduction

Driven by rapid advancements in Earth observation technology, the volume of remote sensing data has witnessed exponential growth (Zhang et al., 2016; Guan et al., 2024b; Huang et al., 2024). Modern remote sensing data is intrinsically characterized by significant multi-source heterogeneity, encompassing diverse modalities such as hyperspectral imagery (HSI), synthetic aperture radar (SAR), and LiDAR point clouds (Shabbir et al., 2025). These modalities capture distinct physical properties of the Earth's surface, which is critical for constructing a holistic understanding of complex terrestrial environments. However, the sheer scale and complexity of this multi-view data render manual interpretation prohibitively expensive, establishing multi-view clustering (MVC) (Huang et al., 2021; Feng et al., 2025; Guo et al., 2025; 2026) in remote sensing as a pivotal technique for mining latent value from unlabeled Earth observation archives (Guan et al., 2024a; Cai et al., 2024; Cao et al., 2025a; Song et al., 2026).

To effectively utilize this heterogeneous data, MVC for remote sensing data has emerged as the dominant unsupervised paradigm. While single-view approaches (Guan et al., 2024b) are limited to the feature space of a specific sensor, often failing to disambiguate complex land cover types with similar spectral but distinct geometric signatures, MVC leverages the synergistic information distributed across distinct views (Liu & Chang, 2025; Guan et al., 2025d; Cao et al., 2025b). By theoretically modeling the consistency and complementarity between views, MVC aims to learn a fused and discriminative latent representation that disentangles complex semantic structures more effectively than any single modality could in isolation. In the context of remote sensing, this capability is indispensable; for instance, fusing the spectral richness of HSI with the structural robustness of LiDAR allows for the precise clustering of land covers that are otherwise indistinguishable (Guan et al., 2026). Consequently, centralized MVC (CMVC) methods have achieved remarkable progress in extracting high-level semantic patterns from multi-modal remote sensing data (Liu & Chang, 2025; Zhang et al., 2025; Liu et al., 2023).

Despite the theoretical efficacy of CMVC, they operate on a centralized assumption, necessitating the aggregation of all data onto a single server (Guan et al., 2025b; Yu et al., 2024). This paradigm is increasingly untenable due to two primary constraints: privacy and efficiency. First, remote sensing data frequently contains sensitive geospatial information, leading to strict data silos where different institutions cannot share raw data directly (Zhu et al., 2023). Second, the communication bandwidth and storage costs required to transmit petabytes of multi-view data to a central node are prohibitive. Federated learning (FL) (Kairouz et al., 2021) offers a promising solution by enabling collaborative learning without raw data transmission (Yu et al., 2026; Meng et al., 2024b). Despite the proliferation of FL in supervised classification tasks (Lin et al., 2025), the domain of federated MVC (FMVC) for remote sensing remains entirely unexplored. This absence is not without reason; establishing a globally consistent clustering model in this setting presents profound technical challenges. Unlike centralized or supervised scenarios, strict privacy constraints prohibit the transmission of raw data or intermediate features to a central server, depriving the system of a unified semantic reference space. Consequently, local models optimized on heterogeneous views in isolation inevitably suffer from severe semantic misalignment. Therefore, solving the multi-view alignment problem solely through parameter communication, without direct feature alignment, stands as a critical and unsolved bottleneck in remote sensing analysis.

To surmount the aforementioned challenges, we propose a deep FMVC framework specifically tailored for remote sensing data, designed to strike an optimal balance between computational efficiency and privacy preservation, as illustrated in Figure 1. We optimize efficiency from two perspectives. First, we employ a superpixel segmentation algorithm to process raw remote sensing imagery, transforming high-dimensional data into compact superpixel units. Second, we transmit only privatized clustering prototypes to the server rather than raw data features, which drastically lowers the bandwidth consumption during the communication phase. Regarding privacy and security, we add noise perturbations to the clustering prototypes to ensure the privacy of the transmission process. Furthermore, we achieve a globally unified clustering result by fusing soft labels generated by distributed clients, effectively preventing the leakage of sensitive latent representations. However, stochastic variations in local clustering processes often lead to misalignment between the privatized prototypes and soft labels generated across different clients. To mitigate this issue, we introduce a co-occurrence structure consistency module to facilitate initial structural alignment. Furthermore, we propose a wasserstein prototype contrastive learning algorithm to enhance the discriminative capability of the model and ensure spatial consistency with the prototypes. The main contributions of this work are summarized as follows.

- **New research task.** We propose the first deep FMVC framework for remote sensing data that effectively balances efficiency and privacy. By utilizing privatized prototypes and soft labels as the core interaction media, we achieve efficient and secure collaboration.

- **Novel FMVC framework.** To ensure consistent global clustering results, we design a co-occurrence structure consistency module and a wasserstein prototype contrastive learning module, which effectively promote the alignment of privatized prototypes uploaded by diverse clients and guarantee clustering performance.

- **Superior empirical performance.** Extensive experiments conducted on four public multi-view remote sensing datasets demonstrate the superior effectiveness and efficiency of the proposed method compared to state-of-the-art baselines.

## 2. Related Work

### 2.1. Deep Multi-view Clustering in Remote Sensing

The increasing prevalence of remote sensing data characterized by multi-modal heterogeneity has rendered traditional single-view clustering methods inadequate for capturing the comprehensive semantic structure of land cover (Cao et al., 2025b). Consequently, research attention has shifted toward MVC (Guan et al., 2025c; Wan et al., 2023; Jin et al., 2023; 2025). Driven by the necessity to extract non-linear and hierarchical features from complex terrain data, the field has evolved from shallow to deep clustering

paradigms (Peng et al., 2025). Existing deep MVC methods for remote sensing data can be broadly categorized into deep autoencoder-based and contrastive learning-based approaches (Shahi et al., 2022). Autoencoder-based methods typically employ an encoder architecture to map heterogeneous views into a common latent space, utilizing a decoder to reconstruct features. For instance, Guan *et al.* (Guan et al., 2025a) proposed a clustering framework with dual-relation optimization, designed to enhance similarity relationships across multi-view data while suppressing noisy correlations. Similarly, Peng *et al.* (Peng et al., 2025) introduced dual-structure-aware MVC, which generates two distinct structures for each view via both explicit and implicit perspectives. Furthermore, SAMVGC (Guan et al., 2025d) adopts a structure-adaptive approach to optimize the graph structure, thereby improving the homogeneity of adjacency relationships. In contrast, contrastive learning-based methods aim to maximize the mutual information between different views of the same scene while minimizing that of negative pairs (Luo et al., 2024; Liu et al., 2025b). Guan *et al.* (Guan et al., 2025b) proposed a sampling-augmented contrastive learning method to improve the quality of positive and negative pairs. Cai *et al.* proposed (Cai et al., 2023) a Transformer-based multi-view prototype contrastive clustering method, which learns a shared feature space through adaptive inter-modal and intra-modal interactions. However, these centralized approaches typically require aggregating massive multi-source datasets onto a single server. This paradigm not only incurs prohibitive storage costs but also disregards the critical privacy constraints associated with sensitive geospatial data.

## 2.2. Federated Multi-View Clustering

FMVC combines distributed privacy-preserving computing with multi-view learning, aiming to fundamentally resolve the problem of data silos (Hu et al., 2024). Existing FMVC methodologies can be broadly categorized into traditional approaches (Liu et al., 2025a) and deep learning-based approaches (Chen et al., 2023). Traditional FMVC approaches primarily focus on extending classical clustering techniques to the FL framework. For instance, Feng *et al.* (Feng et al., 2024a) integrated matrix factorization with multi-view K-means within a federated setting, achieving MVC without sharing raw data. Similarly, Liu *et al.* (Liu et al., 2026) designed a communication-efficient FMVC framework that approximates data representation through the cooperative optimization of pseudo-labels and centroid matrices. In contrast, deep FMVC approaches leverage the powerful representation capabilities of neural networks to perform high-level feature extraction locally on each client (Jiang et al., 2024; Sun et al., 2025). Chao *et al.* (Chao et al., 2025) proposed a framework based on a dual-head Graph Autoencoder (GAE), which introduces a global fusion graph to guide local encoders, thereby enabling the capture of global inter-view correlations and improving feature consistency. Furthermore, Chen *et al.* (Chen et al., 2024) introduced FMCSC, a method designed for federated clustering with heterogeneous hybrid views. They provided rigorous theoretical analysis demonstrating the method's effectiveness in bridging both client-level and view-level heterogeneity. Jiang *et al.* (Jiang et al., 2024) incorporated a heterogeneity-aware module that dynamically assesses the degree of data heterogeneity based on local clustering results. However, these general-domain approaches are not tailored for Earth observation tasks; they typically struggle to handle the massive scale and high dimensionality of remote sensing data, often incurring prohibitive communication and computational overheads in practical deployment.

## 3. Methodology

We propose a privacy-preserving FMVC framework named FedRSMVC tailored for remote sensing data. Our method addresses the challenges of high computational cost and privacy constraints. Figure 2 illustrates the main process framework of FedRSMVC.

### 3.1. Notations and Problem Definition

Let $\mathcal{V} = \{1, 2, \ldots, V\}$ denote the set of $V$ clients participating in the federated system. Each client $v \in \mathcal{V}$ holds a distinct view of the same geographical area, represented by raw imagery $\mathcal{X}^{(v)} \in \mathbb{R}^{H \times W \times D^{(v)}}$, where $H, W$ are spatial dimensions and $D^{(v)}$ is the channel dimension. To reduce computational complexity and establish spatial correspondence, a superpixel segmentation algorithm (Achanta et al., 2012) is executed on the first view $\mathcal{X}^{(1)}$, generating a segmentation map $\mathcal{M} \in \mathbb{R}^{M \times D^{(1)}}$ containing $M$ superpixels. This map $\mathcal{M}$ is shared with all other clients. Each client then aggregates features within each superpixel region to form their local data matrix $\mathbf{X}^{(v)} = [\mathbf{x}_1^{(v)}, \ldots, \mathbf{x}_M^{(v)}]^\top \in \mathbb{R}^{M \times D^{(v)}}$. A crucial property of this setup is that for any two clients $v \neq u$, the samples $x_i^{(v)}$ and $x_i^{(u)}$ correspond to the exact same physical region defined by the $i$-th superpixel in $\mathcal{M}$.

We construct a spatial adjacency matrix $\mathbf{A} \in \{0, 1\}^{M \times M}$ by fingding the $k$ nearest neighbors for each sample $\mathbf{x}^{(v)}$, where $\mathbf{A}_{ij} = 1$ if superpixels $i$ and $j$ are spatially adjacent, and $\mathbf{A}_{ij} = 0$ otherwise. To facilitate graph convolutions, we compute the normalized adjacency matrix with self-loops: $\tilde{\mathbf{A}} = \mathbf{A} + \mathbf{I}_M$, where $\mathbf{I}_M$ is the identity matrix. The degree matrix is $\tilde{\mathbf{D}}_{ii} = \sum_j \tilde{\mathbf{A}}_{ij}$. The final normalized adjacency is $\hat{\mathbf{A}} = \tilde{\mathbf{D}}^{-\frac{1}{2}} \tilde{\mathbf{A}} \tilde{\mathbf{D}}^{-\frac{1}{2}}$. The objective is to partition the $M$ superpixels into $K$ semantic clusters without sharing $\mathbf{X}^{(v)}$ across clients.

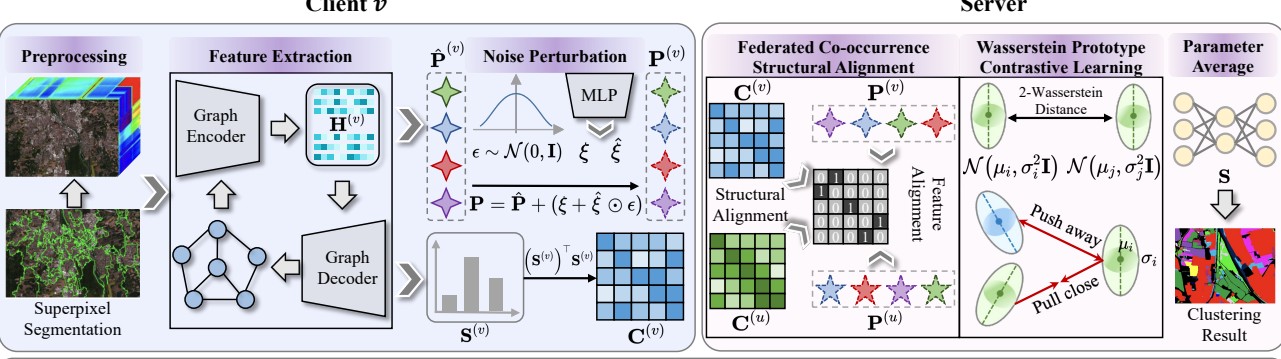

$\mathbf{H}^{(v)}$: Feature of the $v$-client $\mathbf{P}^{(v)}$: Privatized prototypes of the $v$-client $\mathbf{S}^{(v)}$: Assignment matrix of the $v$-client $\mathbf{C}^{(v)}$: Co-occurrence matrix of the $v$-client

*Figure 2.* The overall architecture of the proposed FedRSMVC framework. Left (Client $v$): Raw remote sensing data undergo superpixel segmentation for efficiency. To guarantee privacy, raw prototypes $\hat{\mathbf{P}}^{(v)}$ are perturbed by an adaptive noise mechanism to generate secure prototypes $\mathbf{P}^{(v)}$ for transmission. Right (Server): The server employs two novel modules: (1) Federated Co-occurrence Structural Alignment: utilizes the co-occurrence matrix derived from soft labels to structurally align mismatched prototypes across clients. (2) Wasserstein Prototype Contrastive Learning: model prototypes as distributions $\mathcal{N}(\mu, \Sigma)$ and optimizes the 2-Wasserstein Distance.

## 3.2. Client-side Graph Autoencoder

Each client $v$ locally trains a GAE to map high-dimensional features $\mathbf{X}^{(v)}$ into a low-dimensional latent space $\mathbf{H}^{(v)}$ while preserving spatial contextual information. The encoder $f_{enc}^{(v)}(\cdot; \theta_{enc}^{(v)})$ consists of multiple graph convolutional network layers. The propagation rule for the $l$-th layer is given by:

$$\mathbf{H}^{(v,l+1)} = \text{ReLU}\left(\hat{\mathbf{A}}^{(v)}\mathbf{H}^{(v,l)}\mathbf{W}_{enc}^{(v,l)}\right), \quad (1)$$

where $\mathbf{H}^{(v,0)} = \mathbf{X}^{(v)}$, $\mathbf{W}_{enc}^{(v,l)}$ is the trainable weight matrix of layer $l$. The output of the final encoder layer is the latent representation matrix $\mathbf{H}^{(v)} = [\mathbf{h}_1^{(v)}, \ldots, \mathbf{h}_M^{(v)}]^\top \in \mathbb{R}^{M \times d}$. Unlike standard GAEs that reconstruct the adjacency matrix, our decoder $f_{dec}^{(v)}(\cdot; \theta_{dec}^{(v)})$ aims to reconstruct the node features to ensure the latent space retains semantic content. The decoding process is as follows:

$$\hat{\mathbf{X}}^{(v)} = f_{dec}^{(v)}(\mathbf{H}^{(v)}, \hat{\mathbf{A}}^{(v)}; \theta_{dec}^{(v)}), \quad (2)$$

where $\hat{\mathbf{X}}^{(v)}$ is the reconstructed feature matrix. The local reconstruction loss for client $v$ is defined as the mean squared error:

$$\mathcal{L}_{rec}^{(v)} = \frac{1}{M}\left\|\mathbf{X}^{(v)} - \hat{\mathbf{X}}^{(v)}\right\|_F^2. \quad (3)$$

### 3.2.1. PERTURBATION-BASED SECURITY MODULE

To strictly preserve data privacy within the federated framework, we circumvent the transmission of raw data features. Instead, each client $v$ computes $K$ local prototypes, denoted as $\hat{\mathbf{P}}^{(v)} = \{\hat{\mu}_1^{(v)}, \ldots, \hat{\mu}_K^{(v)}\} \in \mathbb{R}^{K \times d}$, typically derived via local K-means clustering. While communicating abstract prototypes rather than instances reduces immediate privacy

risks, it may still leave the system vulnerable to model inversion or reconstruction attacks. To mitigate this, we employ a stochastic perturbation mechanism to obfuscate the prototypes $\hat{\mathbf{P}}^{(v)}$ before transmission. Specifically, we first sample a noise seed $\epsilon$ from a standard multivariate Gaussian distribution, i.e., $\epsilon \sim \mathcal{N}(0, \mathbf{I})$. To ensure the perturbation adapts to the feature distribution, we estimate a sample-dependent mean $\xi^{(v)}$ and deviation $\hat{\xi}^{(v)}$ using a learnable dual-branch multi-layer perceptron conditioned on the prototypes $\hat{\mathbf{P}}^{(v)}$:

$$\xi^{(v)} = \text{MLP}_\xi^{(v)}(\hat{\mathbf{P}}^{(v)}), \quad \hat{\xi}^{(v)} = \text{MLP}_{\hat{\xi}}^{(v)}(\hat{\mathbf{P}}^{(v)}). \quad (4)$$

Subsequently, the adaptive perturbation $\varepsilon^{(v)}$ is generated via a reparameterization strategy:

$$\varepsilon^{(v)} = \xi^{(v)} + \hat{\xi}^{(v)} \odot \epsilon^{(v)}, \quad (5)$$

where $\odot$ denotes the Hadamard product. The final privacy-preserving prototypes are obtained by injecting this perturbation:

$$\mathbf{P}^{(v)} = \hat{\mathbf{P}}^{(v)} + \varepsilon^{(v)}. \quad (6)$$

Consequently, only the privatized prototypes $\mathbf{P}^{(v)} = \{\mu_1^{(v)}, \ldots, \mu_K^{(v)}\} \in \mathbb{R}^{K \times d}$ is uploaded to the server. This ensures that the exact privatized prototypes $\hat{\mathbf{P}}^{(v)}$ remain concealed, providing robust security guarantees during the communication phase.

## 3.3. Federated Co-occurrence Structural Alignment

Due to the independent initialization and training of local models, the prototypes are subject to arbitrary permutations, i.e., the $i$-th prototype $\mu_i^{(v)}$ in view $v$ does not necessarily correspond to $\mu_j^{(u)}$ in view $u$. We propose Federated Co-occurrence Structural Alignment Model (FCSA) to resolve

this permutation ambiguity by leveraging the inherent alignment of samples. The core insight is that if prototype $i$ in view $v$ corresponds to prototype $j$ in view $u$, they must exhibit similar co-occurrence patterns with respect to the aligned samples. First, each client $v$ computes a soft assignment matrix $\mathbf{S}^{(v)} \in \mathbb{R}^{M \times K}$, representing the probability of each sample belonging to each prototype. To ensure that each data point is assigned to a prototype, we further require that the closest prototype to any given data point has a higher probability of belonging to the same class. Accordingly, for the $v$-th view, the optimization problem of learning the soft assignment matrix can be formulated as follows:

$$\min_{\mathbf{S}^{(v)}} \sum_{i=1}^{M} \sum_{j=1}^{K} \left\| \mathbf{H}_i^{(v)} - \hat{\mathbf{P}}_j^{(v)} \right\|_2^2 s_{ij}^{(v)} + \gamma \left\| \mathbf{S}^{(v)} \right\|_F^2,$$
$$\text{s.t. } \forall i \left\| \mathbf{s}_i^{(v)} \right\|_1 = 1, 0 \leq \mathbf{s}_i^{(v)} \leq 1, \tag{7}$$

where $\mathbf{s}_i^{(v)}$ is the $i$-th row of the soft assignment matrix $\mathbf{S}^{(v)}$ and the sum of each row in $\mathbf{S}^{(v)}$ is 1. The second term ensures that each sample only connects with its nearest prototype. Next, we construct the co-occurrence matrix $\mathbf{C}^{(v)} \in \mathbb{R}^{K \times K}$, defined as the Gram matrix of the assignment matrix:

$$\mathbf{C}^{(v)} = (\mathbf{S}^{(v)})^\top \mathbf{S}^{(v)}. \tag{8}$$

Each element $\mathbf{C}_{ij}^{(v)}$ quantifies the aggregate tendency of prototypes $i$ and $j$ to be simultaneously activated by the same set of samples across the entire dataset. Since the samples on the client are aligned, if a true permutation exists between the prototype sets, the corresponding co-occurrence graphs $\mathbf{C}^{(v)}$ and $\mathbf{C}^{(u)}$ must be topologically isomorphic. Clients upload only their prototypes $\mathbf{P}^{(v)}$ and the co-occurrence matrix $\mathbf{C}^{(v)}$ to the server. Taking views $v$ and $u$ as an example, the server determines the optimal permutation matrix $\mathbf{M}^{(vu)} \in \{0, 1\}^{K \times K}$ that aligns view $u$ to view $v$ by minimizing the structural mismatch between their co-occurrence graphs:

$$\mathcal{L}_{fcsa}^{(v)} =$$
$$\underbrace{\left\| \mathbf{P}^{(v)} - \mathbf{M}^{(vu)} \mathbf{P}^{(u)} \right\|_F^2}_{\text{1. Feature alignment}} + \underbrace{\left\| \mathbf{C}^{(v)} - \mathbf{M}^{(vu)} \mathbf{C}^{(u)} \mathbf{M}^{(vu)^\top} \right\|_F^2}_{\text{2. Structural alignment}}.$$
$$\tag{9}$$

The term $\mathbf{M}^{(vu)} \mathbf{C}^{(u)} \mathbf{M}^{(vu)^\top}$ represents the aligned matrix of $\mathbf{C}^{(u)}$ after reordering its nodes according to $\mathbf{M}^{(vu)}$. To make the optimization of the permutation matrix suitable for the derivation mechanism in deep neural networks, we follow the method in PVC (Huang et al., 2020) and use the differentiable substitution of the Hungarian algorithm to establish the correspondence of prototype sets. Formally

$$\Omega_1 = \text{ReLU}\left( \mathbf{M}^{(vu)} \right), \tag{10}$$

$$\Omega_2 = \mathbf{M}^{(vu)} - \frac{1}{M} \left( \mathbf{M}^{(vu)} \mathbf{1} - \mathbf{1} \right) \mathbf{1}^\top, \tag{11}$$

$$\Omega_3 = \mathbf{M}^{(vu)} - \frac{1}{M} \mathbf{1} \left( \mathbf{1}^\top \mathbf{M}^{(vu)} - \mathbf{1}^\top \right), \tag{12}$$

where $\Omega_1$, $\Omega_2$ and $\Omega_3$ project the permutation matrix $\mathbf{M}^{(vu)}$ to the three constraint sets. Upon obtaining optimized $\tilde{\mathbf{M}}^{(vu)}$, the server computes the structurally aligned prototypes for view $u$ relative to view $v$:

$$\tilde{\mathbf{P}}^{(u \to v)} = \tilde{\mathbf{M}}^{(vu)} \mathbf{P}^{(u)}. \tag{13}$$

Let $\tilde{\mu}_k^{(u \to v)}$ denote the $k$-th row of $\tilde{\mathbf{P}}^{(u \to v)}$. After FCSA, $\mu_k^{(v)}$ and $\tilde{\mu}_k^{(u \to v)}$ are initially semantically aligned.

### 3.4. Wasserstein Prototype Contrastive Learning

While FCSA resolves index permutation, mere alignment of geometric centers is insufficient given the heterogeneity across views. A robust prototype should represent a compact, well-defined cluster. To incorporate distributional information into the alignment, we propose a Wasserstein Prototype Contrastive Learning Model (WPCL). We model each prototype not as a deterministic point, but as an isotropic multivariate Gaussian distribution $\mathcal{N}(\mu_i^{(v)}, (\sigma_i^{(v)})^2 \mathbf{I})$, characterized by its mean center $\mu_i^{(v)}$ and a scalar uncertainty $\sigma_i^{(v)} \in \mathbb{R}^+$. Each client locally estimates the uncertainty $\sigma_i^{(v)}$ for each prototype, defined as the weighted average Euclidean distance of assigned samples from the cluster center:

$$\sigma_i^{(v)} = \sqrt{\frac{\sum_{j=1}^{M} \mathbf{S}_{ji}^{(v)} \left\| \mathbf{h}_j^{(v)} - \mu_i^{(v)} \right\|_2^2}{\sum_{j=1}^{M} \mathbf{S}_{ji}^{(v)} + \eta}} \tag{14}$$

where $\eta$ is a small constant for numerical stability. To measure the difference between two prototype distributions $\mathbf{P}_i^{(v)}$ and $\mathbf{P}_j^{(u)}$ in different views, we use the 2-Wasserstein distance. For two general multivariate Gaussian distributions $\mathcal{N}(\mu_i^{(v)}, \mathbf{\Sigma}_i^{(v)})$ and $\mathcal{N}(\mu_j^{(u)}, \mathbf{\Sigma}_j^{(u)})$, the squared 2-Wasserstein distance between them has the following closed-form solution:

$$W_2^2 \left( \mathcal{N}_i^{(v)}, \mathcal{N}_j^{(u)} \right) = \left\| \mu_i^{(v)} - \mu_j^{(u)} \right\|_2^2 +$$
$$\text{Tr} \left( \mathbf{\Sigma}_i^{(v)} + \mathbf{\Sigma}_j^{(u)} - 2 \left( (\mathbf{\Sigma}_i^{(v)})^{1/2} \mathbf{\Sigma}_j^{(u)} (\mathbf{\Sigma}_i^{(v)})^{1/2} \right)^{1/2} \right). \tag{15}$$

Under our isotropic assumption, the covariance matrix simplifies to $\mathbf{\Sigma}_i^{(v)} = (\sigma_i^{(v)})^2 \mathbf{I}$ and $\mathbf{\Sigma}_j^{(u)} = (\sigma_j^{(u)})^2 \mathbf{I}$. Substituting these into Eq. (15), the trajectory term can be simplified

to:

$$
\begin{aligned}
&\text{Tr} \left( \boldsymbol{\Sigma}_i^{(v)} + \boldsymbol{\Sigma}_j^{(u)} - 2 \left( (\boldsymbol{\Sigma}_i^{(v)})^{1/2} \boldsymbol{\Sigma}_j^{(u)} (\boldsymbol{\Sigma}_i^{(v)})^{1/2} \right)^{1/2} \right) \\
&= \text{Tr} \left( \left( \left( \sigma_i^{(v)} \right)^2 + \left( \sigma_j^{(u)} \right)^2 - 2\sigma_i^{(v)} \sigma_j^{(u)} \right) \mathbf{I} \right) \\
&= \text{Tr} \left( \left( \sigma_i^{(v)} - \sigma_j^{(u)} \right)^2 \mathbf{I} \right) \\
&= d \left( \sigma_i^{(v)} - \sigma_j^{(u)} \right)^2,
\end{aligned}
$$
(16)

where $d$ is the feature dimension and $\mathbf{I} \in \mathbb{R}^{d \times d}$ is the identity matrix. Therefore, the squared 2-Wasserstein distance between the two prototype distributions $\mathbf{P}_i^{(v)}$ and $\mathbf{P}_j^{(u)}$ can be expressed as:

$$
\begin{aligned}
&\mathcal{S}_W(\mathbf{P}_i^{(v)}, \mathbf{P}_j^{(u \to v)}) = \\
&- \left( \left\| \mu_i^{(v)} - \tilde{\mu}_j^{(u \to v)} \right\|_2^2 + \lambda (\sigma_i^{(v)} - \tilde{\sigma}_j^{(u \to v)})^2 \right),
\end{aligned}
$$
(17)

where $\lambda > 0$ is a hyperparameter used to balance the importance of center alignment and uncertain alignment, and it absorbs the influence of the feature dimension $d$. Finally, we incorporate this similarity metric into the InfoNCE loss framework. For a prototype $i$ of view $v$, its positive sample in view $u$ is prototype $i$, while the other $K-1$ prototypes are considered negative samples. The Wasserstein prototype contrastive loss from view $v$ to view $u$ is defined as:

$$
\mathcal{L}_{wpcl}^{(v)} = -\sum_{i=1}^{K} \log \frac{\exp \left( \mathcal{S}_W \left( \mathbf{P}_i^{(v)}, \mathbf{P}_i^{(u \to v)} \right) / \tau \right)}{\sum_{k=1}^{K} \exp \left( \mathcal{S}_W \left( \mathbf{P}_i^{(v)}, \mathbf{P}_k^{(u \to v)} \right) / \tau \right)},
$$
(18)

where $\tau$ is the temperature parameter. By minimizing this loss function, the model is encouraged to bring the distribution of corresponding prototypes closer together in the Wasserstein geometric space, while pushing away the distribution of non-corresponding prototypes, thereby achieving more rigorous and reliable cross-view prototype alignment.

**Theorem 1.** Let $\mathbf{P}^{(v)}$ and $\mathbf{P}^{(u)}$ denote the random variables representing prototypes from two distinct views. The proposed WPCL loss $\mathcal{L}_{wpcl}$, satisfies the following inequality chain:

$$
-\mathcal{L}_{wpcl} \leq I_{NCE} \leq I(\mathbf{P}^{(v)}; \mathbf{P}^{(u)}),
$$
(19)

where $I_{NCE}$ represents the supremum of the InfoNCE lower bound over the space of unconstrained critic functions.

### 3.5. Total Loss and Optimization

The overall objective is to jointly optimize local feature representation, global prototype alignment and wasserstein prototype contrastive learning. The total loss function for client $v$ is

$$
\mathcal{L}^{(v)} = \mathcal{L}_{rec}^{(v)} + \lambda_1 \sum_{v=1}^{V} \mathcal{L}_{fcsa}^{(v)} + \lambda_2 \sum_{v=1}^{V} \mathcal{L}_{wpcl}^{(v)},
$$
(20)

where $\lambda_1$ and $\lambda_2$ are hyperparameters for balancing loss $\mathcal{L}_{fcsa}$ and $\mathcal{L}_{wpcl}$, respectively. The alignment loss $\mathcal{L}_{fcsa}^{(v)}$ and the contrastive loss $\mathcal{L}_{wpcl}$ are computed on the server, and only the corresponding gradients are transmitted back to each client. Each client then jointly optimizes local model using these gradients together with the reconstruction loss $\mathcal{L}_{rec}^{(v)}$, thereby preserving data privacy and ensuring secure communication. After model optimization, we aggregate the aligned soft assignment matrix and output the hard labels $\mathbf{S}$ as the final unified result:

$$
\mathbf{S} = \text{Softmax} \left( \frac{1}{V} \sum_{v=1}^{V} \mathbf{S}^{(v)} \right).
$$
(21)

## 4. Experiments

### 4.1. Experimental Settings

**Datasets.** To comprehensively evaluate the performance of FedRSMVC, we conduct experiments on four widely-used multi-view remote sensing benchmarks: Trento, XuZhou, Augsburg, and Salinas. These datasets exhibit diverse characteristics in terms of sample size, class distribution, and feature modalities. Specifically, the Trento dataset comprises Hyperspectral (HS) and LiDAR data with 6 distinct clusters. The Augsburg dataset integrates HS, SAR, and DSM views, offering a challenging multimodal scenario. XuZhou and Salinas datasets both combine HS imagery with Extended Morphological Profiles (EMP) and 3D Gabor features, containing 9 and 16 classes, respectively.

**Compared Methods.** To fully compare the advantages of FedRSMVC, we chose to compare it with nine FMVC methods. These methods are FedMVL (Huang et al., 2022), FedDMVC (Chen et al., 2023), FMVC-IMK (Feng et al., 2024a), TensorFMVC (Feng et al., 2024b), Fed-MVFCM (Hu et al., 2024), FedMVFPC (Hu et al., 2024), MGCD (Sun et al., 2025), CeFMC (Liu et al., 2026), Fed-MVKM (Yang & Sinaga, 2025).

**Implement Detail.** We implement FedRSMVC using the PyTorch framework on a workstation equipped with a single 24GB NVIDIA GeForce RTX 3090 GPU and 64GB RAM. The model is optimized using the Adam optimizer with a learning rate of $10^{-3}$ and a weight decay of $10^{-5}$. The number of communication rounds is set to $T = 200$. For the loss balancing hyperparameters, we set $\lambda_1 = 1$ and $\lambda_2 = 1$. To ensure statistical reliability, all reported results represent the average of 10 independent trials (Zhuo et al., 2024b;a; Yan et al., 2025; Meng et al., 2024a; 2026).

*Table 1.* Quantitative clustering performance comparison (mean ± standard deviation) on four multi-view remote sensing benchmarks.

| Method | Trento | | | | | Salinas | | | | |
|---|---|---|---|---|---|---|---|---|---|---|
| | ACC | KAPPA | NMI | ARI | PUR | ACC | KAPPA | NMI | ARI | PUR |
| **FedMVL** PR'22 | 33.9±1.6 | 11.5±2.2 | 5.4±2.3 | 4.3±1.7 | 43.9±2.5 | 41.5±4.6 | 36.4±5.0 | 48.0±7.7 | 28.6±6.4 | 43.1±5.6 |
| **FedDMVC** MM'23 | 71.7±2.0 | 61.0±2.8 | 64.1±4.9 | 58.0±6.0 | 74.3±2.2 | 53.7±1.3 | 42.8±2.5 | 70.2±2.4 | 41.7±2.3 | 46.1±3.5 |
| **FMVC-IMK** IJCAI'24 | | | OOM | | | | | OOM | | |
| **TensorFMVC** IJCAI'24 | | | OOM | | | | | OOM | | |
| **Fed-MVFCM** TFS'24 | 47.6±0.8 | 17.9±5.1 | 44.4±0.3 | 31.5±0.7 | 64.3±0.0 | 50.6±1.1 | 43.0±2.0 | 45.2±1.1 | 36.3±0.4 | 43.0±2.1 |
| **Fed-MVFPC** TFS'24 | 55.2±2.6 | 12.7±6.8 | 51.4±3.4 | 38.0±3.2 | 77.0±0.0 | 44.9±2.2 | 38.5±3.6 | 59.5±1.1 | 37.0±2.2 | 46.7±2.1 |
| **CeFMC** TPAMI'25 | 39.6±0.2 | 25.0±0.3 | 32.4±0.0 | 17.8±0.0 | 51.7±0.2 | 44.1±6.7 | 36.5±9.5 | 46.8±10.9 | 21.7±5.0 | 48.3±6.5 |
| **MGCD** AAAI'25 | | | OOM | | | | | OOM | | |
| **Fed-MVKM** TPAMI'25 | 60.6±0.0 | 45.8±0.0 | 45.2±0.0 | 36.3±0.0 | 63.0±0.0 | 53.2±3.5 | 47.5±3.8 | 65.2±3.5 | 42.0±5.4 | 56.7±3.7 |
| **FedRSMVC** | **87.0±1.2** | **83.2±1.4** | **77.0±3.8** | **81.6±4.8** | **87.0±1.2** | **73.3±0.9** | **70.4±1.1** | **74.9±1.5** | **64.5±1.6** | **74.3±1.8** |

| Method | XuZhou | | | | | Augsburg | | | | |
|---|---|---|---|---|---|---|---|---|---|---|
| | ACC | KAPPA | NMI | ARI | PUR | ACC | KAPPA | NMI | ARI | PUR |
| **FedMVL** PR'22 | | | OOM | | | | | OOM | | |
| **FedDMVC** MM'23 | 59.7±5.2 | **54.0±9.2** | **54.5±4.3** | **49.9±9.9** | 51.5±6.2 | 46.6±2.3 | 31.2±1.8 | 23.4±2.2 | 18.0±1.1 | 41.8±1.6 |
| **FMVC-IMK** IJCAI'24 | | | OOM | | | | | OOM | | |
| **TensorFMVC** IJCAI'24 | | | OOM | | | | | OOM | | |
| **Fed-MVFCM** TFS'24 | 48.2±5.4 | 27.5±4.2 | 43.2±3.7 | 35.1±9.3 | 59.4±4.4 | 35.1±0.0 | 22.5±0.0 | 23.5±0.0 | 15.1±0.0 | 35.8± 0.1 |
| **Fed-MVFPC** TFS'24 | 53.2±0.2 | 22.3±2.8 | 40.3±0.6 | 43.5±0.1 | 58.1±0.2 | 43.0±0.4 | 30.1±0.1 | 15.2±0.0 | 11.8± 0.2 | 45.1±0.1 |
| **CeFMC** TPAMI'25 | 35.5±2.2 | 24.3±2.3 | 27.6±2.8 | 13.9±1.9 | 46.9±1.8 | 36.9±1.3 | 6.1±2.6 | 13.1±3.4 | 8.5±0.7 | 47.0±3.1 |
| **MGCD** AAAI'25 | | | OOM | | | | | OOM | | |
| **Fed-MVKM** TPAMI'25 | 53.2±5.8 | 42.4±6.4 | 48.0±3.5 | 45.4±7.2 | 62.8±2.7 | 42.1±3.4 | 21.0±3.2 | 16.4±1.1 | 11.1±2.2 | 48.5±2.9 |
| **FedRSMVC** | **61.0±4.0** | 52.3±3.7 | 51.5±0.5 | 48.7±0.9 | **64.8±1.3** | **70.1±0.9** | **56.7 ±1.2** | **44.1±1.5** | **39.8±0.7** | **68.9±1.8** |

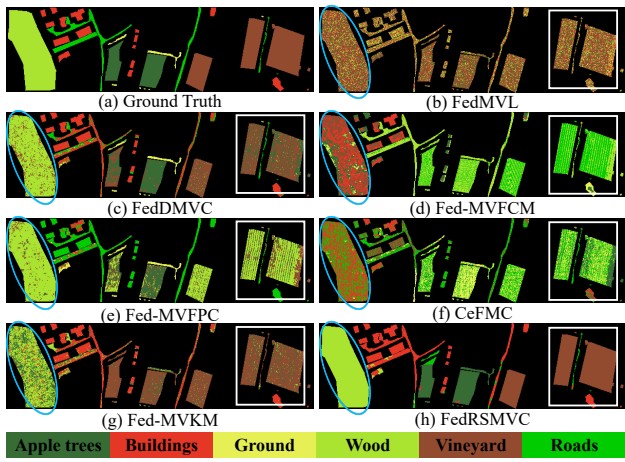

(a) Ground Truth    (b) FedMVL

(c) FedDMVC    (d) Fed-MVFCM

(e) Fed-MVFPC    (f) CeFMC

(g) Fed-MVKM    (h) FedRSMVC

Apple trees | Buildings | Ground | Wood | Vineyard | Roads

*Figure 3.* Visualization of clustering results on the Trento dataset.

*Table 2.* Runtime comparison (in seconds) on four remote sensing datasets.

| Methods | Trento | Salinas | XuZhou | Augsburg |
|---|---|---|---|---|
| FedMVL | 2565.01 | 2269.77 | OOM | OOM |
| FedDMVC | 2565.01 | 2255.77 | 2886.92 | 1588.2 |
| Fed-MVFCM | 34.47 | 182.63 | 365.86 | 438.33 |
| Fed-MVFPC | 1.9 | 24.69 | 28.91 | 27.92 |
| CeFMC | 3.57 | 19.79 | 11.94 | 9.07 |
| Fed-MVKM | 32.48 | 55.36 | 87.6 | 222.21 |
| Ours | 240.19 | 383.96 | 323.24 | 374.03 |

## 4.2. Clustering Results

Table 1 reports the quantitative clustering performance of FedRSMVC against nine SOTA FMVC baselines across four remote sensing benchmarks. The results reveal several critical insights:

**(i) Superiority over federated baselines.** FedRSMVC consistently achieves the highest clustering accuracy among all federated approaches, establishing a new state-of-the-art. On the Trento dataset, our method outperforms the second-best federated competitor, FedDMVC, by a substantial margin of 15.3% in ACC and 12.9% in NMI. Similarly, on the challenging Augsburg dataset, where multi-modal heterogeneity complicates feature alignment, FedRSMVC surpasses the best performing federated baseline (Fed-MVKM) by 7.9% in ACC and 26.9% in NMI.

**(ii) Trade-off between speed and performance.** As demonstrated in Table 2, we observe that traditional FMVC methods, such as CeFMC and Fed-MVFPC, achieve lower runtime values. However, this speed comes at the cost of representation capability. In contrast, FedRSMVC achieves the best trade-off: it is the fastest among all deep learning-based methods, providing SOTA clustering performance with a runtime comparable to traditional algorithms on larger datasets.

**(iii) Visualization analysis.** To intuitively evaluate clustering quality, Figure 3 visualizes the clustering results on

*Table 3.* Ablation study of key components on four multi-view remote sensing datasets.

| Variants | | | Trento | | | | Salinas | | | | XuZhou | | | | Augsburg | | | |
|---|---|---|---|---|---|---|---|---|---|---|---|---|---|---|---|---|---|---|
| $\mathcal{L}_{rec}$ | $\mathcal{L}_{fcsa}$ | $\mathcal{L}_{wpcl}$ | OA | Kappa | NMI | ARI | OA | Kappa | NMI | ARI | OA | Kappa | NMI | ARI | OA | Kappa | NMI | ARI |
| ✔ | ✘ | ✘ | 56.3 | 42.7 | 44.4 | 33.4 | 37.9 | 27.4 | 43.7 | 25.9 | 46.1 | 32.5 | 33.4 | 23.1 | 53.1 | 23.1 | 19.5 | 22.7 |
| ✔ | ✔ | ✘ | 61.8 | 46.9 | 50.6 | 39.4 | 41.5 | 32.7 | 52.4 | 29.7 | 49.5 | 35.6 | 34.1 | 25.8 | 60.3 | 39.8 | 30.8 | 29.4 |
| ✔ | ✘ | ✔ | 66.9 | 56.2 | 55.2 | 50.2 | 41.3 | 30.2 | 44.9 | 26.0 | 50.9 | 36.4 | 34.7 | 26.4 | 59.3 | 36.7 | 28.4 | 26.6 |
| ✘ | ✔ | ✔ | 77.4 | 74.9 | 65.9 | 66.4 | 63.2 | 54.4 | 61.6 | 50.2 | 55.2 | 42.3 | 43.0 | 34.6 | 64.3 | 45.0 | 32.8 | 33.0 |
| ✔ | ✔ | ✔ | **87.0** | **83.2** | **77.0** | **81.6** | **73.7** | **70.4** | **74.2** | **64.5** | **61.0** | **52.3** | **51.5** | **48.7** | **70.1** | **56.7** | **44.1** | **39.8** |

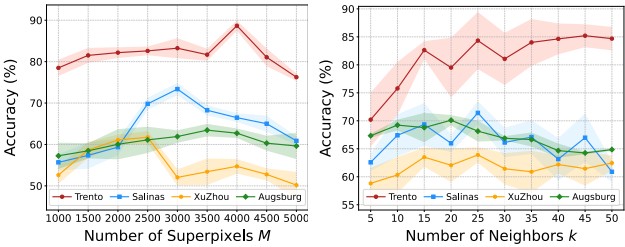

*Figure 4.* Parameter sensitivity analysis regarding the number of superpixels $M$.

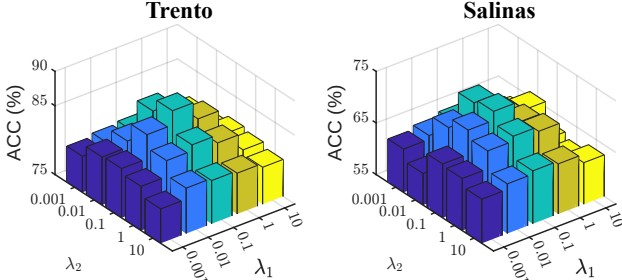

*Figure 5.* Joint sensitivity analysis of the loss balancing hyperparameters $\lambda_1$ and $\lambda_2$ on the Trento and Salinas datasets.

the Trento dataset. In contrast to traditional baselines that exhibit severe salt-and-pepper noise due to the lack of spatial constraints, FedRSMVC achieves the highest visual fidelity. By synergizing superpixel-based spatial smoothing, our method effectively eliminates high-frequency artifacts and corrects semantic ambiguities, yielding a coherent clustering map that closely aligns with the ground truth.

### 4.3. Ablation Study

To rigorously verify the specific contribution of each component within the FedRSMVC framework, we conducted a comprehensive ablation study on all four benchmark datasets. As detailed in Table 3, we constructed four distinct model variants to isolate the effects of the FCSA module and the WPCL module.

The introduction of either $\mathcal{L}_{fcsa}$ or $\mathcal{L}_{wpcl}$ yields substantial performance gains, confirming their effectiveness in resolving specific alignment bottlenecks. Incorporating the FCSA module improves the ACC on Trento by 5.5% (from 56.3% to 61.8%), validating its ability to rectify discrete index mismatches via co-occurrence patterns. Meanwhile, the WPCL module demonstrates a remarkable capacity for handling feature heterogeneity, particularly on the Salinas dataset where it boosts ACC by 17.5% over the baseline. This indicates that modeling prototypes as Gaussian distributions with learnable uncertainty enables the model to effectively push semantically similar clusters together in the latent space, even in the presence of noise. Most importantly, the complete FedRSMVC framework achieves the highest performance metrics across all datasets, demon-

strating a strong synergistic effect between the two modules. The former establishes the necessary global correspondence, while the latter refines the local discriminability, making both indispensable for robust federated clustering.

### 4.4. Parameters Sensitivity Analysis

In this subsection, we conduct a rigorous sensitivity analysis to investigate how the granularity of spatial segmentation and the balancing of loss terms influence the clustering performance of FedRSMVC.

**(i) Impact of superpixel granularity.** The number of superpixels $M$ dictates the granularity of the spatial graph construction. As illustrated in Figure 4, the ACC exhibits a characteristic inverted U-shaped trend across all datasets. In the low-granularity regime, insufficient segmentation leads to mixed pixels where distinct land cover classes are merged into single nodes. Conversely, excessive over-segmentation fragments meaningful objects into redundant nodes, disrupting the long-range structural dependencies required by the GAE. Consequently, we empirically select the optimal $M$ to align with the intrinsic scale of geospatial objects in each dataset.

**(ii) Impact of loss balancing hyperparameters.** We further analyze the sensitivity of hyperparameters $\lambda_1$ and $\lambda_2$, which control the contribution of the FCSA and WPCL modules, respectively. Figure 5 visualizes the joint impact of these terms. Performance drops significantly when these weights approach zero, confirming that relying solely

on local reconstruction is insufficient to resolve prototype misalignment. However, excessively large values lead to over-regularization, where global alignment suppress the learning of discriminative local features.

**(iii) Sensitivity to topological sparsity.** The parameter $k$ plays a pivotal role in constructing the adjacency matrix $\mathbf{A}$, determining the receptive field for the GAE. Figure 4 illustrates the variation in ACC as $k$ ranges from 5 to 50. In the regime of small $k$, the constructed graph is overly sparse, leading to a fragmented manifold where semantic information fails to propagate effectively to intra-class nodes. As $k$ increases, the performance improves significantly, reaching a peak. This suggests that a moderate neighborhood size effectively captures the local manifold structure, facilitating the aggregation of homophilous features.

## 5. Conclusion

In this paper, we presented FedRSMVC, a pioneering framework that extends deep MVC into the FL paradigm, specifically tailored to address the unique challenges of remote sensing data. By introducing FCSA and WPCL module, our approach successfully resolves feature misalignment and models cluster uncertainty across distributed clients without requiring raw data transmission. Extensive experiments on four benchmarks demonstrate that FedRSMVC not only establishes a new state-of-the-art among federated methods but also achieves clustering performance and computational efficiency comparable to centralized baselines.

## Acknowledgment

This work is supported by the National Science Fund for Distinguished Young Scholars of China (No. 62325604), the Major Program Project of Xiangjiang Laboratory (No. 24XJJCYJ01002), the Provincial Natural Science Foundation of Hunan (No. 2025JJ10008) and the National Natural Science Foundation of China (No. 62441618, 62506371, 62276271, 62572480, 62406329 and 62476280).

## Impact Statement

This paper presents work whose goal is to advance the field of Machine Learning. There are many potential societal consequences of our work, none which we feel must be specifically highlighted here.

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
