# OpenReview forum: "Federated Multi-view Clustering for Remote Sensing Data"
_ICML.cc/2026/Conference — ICML 2026 regular_

### Official Review · Reviewer_PAab · 2026-03-01

**Soundness:** 4
**Presentation:** 3
**Significance:** 3
**Originality:** 3
**Overall Recommendation:** 5
**Confidence:** 4

**Summary:**

This paper investigates federated multi view clustering in remote sensing scenarios and proposes a deep federated multi view clustering framework termed FedRSMVC. Motivated by practical deployment settings, the authors point out that multi modal remote sensing data are often distributed across multiple institutions, where privacy constraints and limited communication bandwidth make conventional centralized multi view clustering approaches impractical. To address this issue, superpixel segmentation is applied to reduce spatial dimensionality, and local clustering prototypes are generated and perturbed with adaptive noise before being transmitted to the server, thereby avoiding the sharing of raw features. To resolve the prototype index inconsistency across clients, the authors design a co occurrence matrix based structural alignment module. Furthermore, prototypes are modeled as Gaussian distributions with uncertainty, and a prototype level contrastive learning mechanism based on the second order Wasserstein distance is introduced to perform cross view alignment in the distribution space. Experimental results on four multi view remote sensing datasets demonstrate that the proposed method consistently outperforms existing federated multi view clustering baselines across multiple clustering metrics, while achieving a reasonable trade off between performance and efficiency.

**Compliance With Llm Reviewing Policy:**

Affirmed.

**Key Questions For Authors:**

1. Does the co occurrence based structural alignment require pairwise permutation estimation among all clients, and if so, how does the alignment cost scale with an increasing number of clients.

2. Does the framework assume that all clients share an identical superpixel partitioning scheme, and if data originate from different geographic regions or sensing conditions, would this assumption still hold in practice. Addressing these issues would further strengthen the technical soundness and practical applicability of the proposed framework.

**Limitations:**

Yes

**Strengths And Weaknesses:**

Strengths
1. The problem setting is timely and practically relevant. Extending multi view remote sensing clustering to the federated learning paradigm addresses the increasingly critical challenges of data silos and privacy preservation.

2. The methodological design is coherent and well structured. Rather than directly adapting existing federated clustering frameworks, the authors explicitly tackle the prototype misalignment issue through a co occurrence based structural alignment mechanism and further incorporate Wasserstein geometry to capture distributional discrepancies, resulting in complementary and mutually reinforcing components.

3. The paper provides theoretical derivations for the soft assignment matrix and presents a lower bound analysis based on InfoNCE, which strengthens the formal grounding of the approach.

4. The experimental evaluation is relatively comprehensive, including comparisons on multiple datasets, ablation studies, and parameter sensitivity analyses, and the empirical results convincingly support the effectiveness of the proposed modules.

Weakness

1. Although a large number of baselines are included, the performance gap between the proposed federated approach and strong centralized upper bound methods is not systematically analyzed. It therefore remains unclear how far the method is from the attainable performance ceiling under privacy constraints.

2. While computational complexity is discussed in the appendix, the analysis lacks explicit asymptotic characterization with respect to the number of communication rounds, the number of prototypes, and the superpixel scale, making it difficult to assess scalability in large scale remote sensing applications.

3. The claimed advantages of Wasserstein based contrastive learning over standard prototype level contrastive losses would benefit from more direct empirical evidence, such as targeted ablations or visualization analyses in the representation space.

---

> ### Author Rebuttal · Authors · 2026-03-25
>
> We sincerely thank you for the constructive feedback and insightful questions. We address your specific concerns in detail below.
>
> **W1:** Regarding the performance gap between our federated and centralized methods, we provide a systematic comparison in Appendix I, point 2, and Table 5 of the Supplementary Material. Results show that our method meets or even surpasses some centralized MVC baselines. For example, on the Augsburg dataset, our framework outperforms SOTA conditional double diffusion methods by 15.8%. We attribute this advantage to the inherent information bottleneck effect in our privacy-preserving mechanism. By transmitting only privatized prototype and structural co-occurrence matrices, the model acts as a learnable filter, discarding redundant pixel-level noise while preserving robust semantic topology.
>
> **W2:** Concerning the computational complexity and scalability, we detailed the theoretical analysis in Appendix F. To provide explicit asymptotic characterization per communication round, the server side complexity scales proportionally to the square of the client count and the cube of the prototype count. Crucially, this server cost is entirely independent of the superpixel scale, ensuring high scalability for massive Earth observation archives. The client side complexity scales linearly with the superpixel scale and the number of GCN layers. The total communication overhead per round scales linearly with the client count and the product of the prototype count and latent feature dimension.
>
> **W3:**
> We entirely agree that providing direct empirical evidence comparing our Wasserstein-based approach to standard prototype-level contrastive losses significantly strengthens the claims of the paper. To address this, we have conducted the targeted ablation study you suggested across all four benchmark datasets. We compared three variants of our framework:
> w/o L_wpcl: The baseline model without any prototype contrastive learning.
>
> w InfoNCE: The model utilizing a standard point-estimation prototype contrastive loss (standard InfoNCE).
>
> w L_wpcl: Our proposed WPCL.
>
> | Variants |  | Trento |  |  |  | Salinas |  |  |  | XuZhou |  |  |  | Augsburg |  |  |
> |----------|--|--|--|--|--|--|--|--|--|--|--|--|--|--|--|--|
> |          | OA | KAPPA | NMI | ARI | OA | KAPPA | NMI | ARI | OA | KAPPA | NMI | ARI | OA | KAPPA | NMI | ARI |
> | w/o L_wpcl | 61.8 | 46.9 | 50.6 | 39.4 | 41.5 | 32.7 | 52.4 | 29.7 | 49.5 | 35.6 | 34.1 | 25.8 | 60.3 | 39.8 | 30.8 | 29.4 |
> | w InfoNce | 79.3 | 75.7 | 70.2 | 73.5 | 57.2 | 53.2 | 68.2 | 46.6 | 55.8 | 42.9 | 44.7 | 41.5 | 66.5 | 50.3 | 38.7 | 34.4 |
> | w L_wpcl | 87.0 | 83.2 | 77.0 | 81.6 | 73.7 | 70.4 | 74.2 | 64.5 | 61.0 | 52.3 | 51.5 | 48.7 | 70.1 | 56.7 | 44.1 | 39.8 |
>
> The empirical evidence demonstrates that while incorporating a standard InfoNCE loss improves performance over the baseline, our proposed WPCL achieves a massive further leap to 73.7% OA. We observe similarly substantial improvements across all other datasets. These performance gaps fundamentally stem from how standard contrastive losses struggle with the extreme heterogeneity of multi-modal data. Standard InfoNCE treats prototypes as deterministic points, entirely ignoring the inherent intra-cluster variance. In contrast, our WPCL leverages optimal transport principles by modeling each prototype not as a deterministic point, but as an isotropic multivariate Gaussian distribution. By utilizing the 2-Wasserstein distance to measure distribution discrepancies, WPCL explicitly incorporates a scalar uncertainty representing the compactness of the cluster. This allows the model to dynamically gauge the reliability of distributions across highly heterogeneous views, thereby pushing semantically similar clusters together far more rigorously than point-based distance metrics.
>
> **R1:** As detailed in Appendix F, performing pairwise alignment across all clients does scale quadratically with the client count. However, in practical federated deployments, this cost is mitigated by establishing a star topology where all clients align their prototypes exclusively to a designated central anchor view, thereby reducing the alignment scaling to a linear relationship with the client count.
>
> **R2:** Our framework indeed operates under the assumption that all clients hold spatially co registered views of the exact same geographical area. As outlined in Section 3 point 1 and Appendix E, we designate a primary modality such as the hyperspectral imagery as the anchor view to execute the Simple Linear Iterative Clustering algorithm. The resulting spatial partition map is then broadcast to all distributed clients to govern their local feature pooling, ensuring strict physical correspondence across heterogeneous modalities. While this assumption is a standard prerequisite in multi modal remote sensing fusion tasks, extending the framework to handle loosely aligned geographic regions remains an important direction for future research.

---

> > ### Author Rebuttal · Reviewer_PAab · 2026-04-02
> >
> > The rebuttal addressed my concerns.

---

### Official Review · Reviewer_wFy9 · 2026-03-06

**Soundness:** 3
**Presentation:** 3
**Significance:** 2
**Originality:** 2
**Overall Recommendation:** 4
**Confidence:** 3

**Summary:**

The paper proposes a federated multi-view clustering approach with focus on obtaining optimal tradeoff between efficiency and privacy for remote sensing data. The method first transforms the data to superpixels segmentation map which is shared with all clients. All clients then generate local data matrix depending on the shared superpixel map. A graph autoencoder then projects the high dimensional data to low dimensional space while preserving spatial context. These local prototypes are then perturbed using multivariate Gaussian distributions. At server side, structural alignment of these perturbed prototypes is performed and Wasserstein Prototype Contrastive Learning is utilized for geometric alignment of the clusters.
The method provide significant improvement on three datasets in terms of all metrics while marginal improvement in terms of accuracy on 4th dataset while having low KAPPA, NMI and ARI.

**Compliance With Llm Reviewing Policy:**

Affirmed.

**Final Justification:**

My concerns were resolved in rebuttal and i updated my ratings. Therefore, i would like to accept the paper.

**Key Questions For Authors:**

Superpixel map (M) of first view may not be aligned with other views on other clients due to modality specific information. Aggregating the pixels based on M explicitly forces the client views to follow map M while losing the precise information of client views.

Why the proposed method has huge difference in score for Salinas, Augsburg and Trento , while only marginal improvement in terms of accuracy for XuZhou with less KAPPA?

Fig 3 demonstrates that existing methods provide bad results due to the use of multivariate Gaussian noise and may not support the noisy prototypes. It is better to also include their performance with the inputs what these methods support. Also, since the proposed method loss is specifically designed for the noisy prototypes therefore, its performance improvement should be obvious.

Fig 3 should also include the remote sensing multisensors views for better analysis.

Table 1 should replace the OOM with the results on low resolution inputs for better comparison.

**Limitations:**

yes

**Strengths And Weaknesses:**

Strengths:
Proposed method provides decent runtime while providing significant performance improvement.

Weaknesses:
- Superpixel map (M) of first view may not be aligned with other views on other clients due to modality specific information. Aggregating the pixels based on M explicitly forces the client view to follow map M while losing the precise information of client views. Additionally, all the clients already follow the map of first view, then why structural alignment is necessary here?

- Why the proposed method has huge difference in score for Salinas, Augsburg and Trento , while only marginal improvement in terms of accuracy for XuZhou with less KAPPA?

- Fig 3 demonstrates that existing methods provide bad results due to the use of multivariate Gaussian noise and may not support the noisy prototypes. It is better to also include their performance with the inputs what these methods support. Also, since the proposed method loss is specifically designed for the noisy prototypes therefore, its performance improvement should be obvious.

- Fig 3 should also include the remote sensing multisensors views for better analysis.

- Table 1 should replace the OOM with the results on low resolution inputs for better comparison.

---

> ### Author Rebuttal · Authors · 2026-03-30
>
> We sincerely thank you for the constructive feedback and insightful questions. We address your specific concerns in detail below.
>
> **Q1:** In the context of our multi-modal remote sensing framework, the imagery is strictly co-registered spatially, meaning samples $x_{i}^{(v)}$ and $x_{i}^{(u)}$ correspond to the exact same physical region on the Earth's surface. To empirically investigate whether aggregating pixels based on the primary view's map $\mathcal{M}$ causes destructive information loss, we conducted a rigorous quantitative evaluation of the superpixel segmentation purity across all views. Purity measures the percentage of raw pixels within a generated superpixel that share the identical ground-truth semantic label. A high purity indicates that the superpixel successfully encapsulates a homogeneous physical region without crossing semantic boundaries. As shown in the table below (which we will include in the revised Appendix), the empirical purity scores across all datasets and modalities are exceptionally high:
>
> | Dataset  | View 1 | View 2 | View 3 |
> |----------|--------|--------|--------|
> | Trento   | 99.52% | 99.68% | -      |
> | Salinas  | 99.84% | 99.81% | 99.00% |
> | XuZhou   | 99.33% | 99.59% | 99.04% |
> | Augsburg | 99.12% | 99.38% | 99.37% |
>
> Because we designate the first view $\mathcal{X}^{(1)}$ as the anchor view to generate the shared spatial partition map, the generated map $\mathcal{M}$ captures the definitive physical boundaries of land-cover objects. The empirical data proves that these boundaries remain fundamentally consistent across the strictly co-registered sensors.
>
> **Q2:** Regarding the marginal ACC improvement on XuZhou compared to the massive gains on Trento and Augsburg: Trento and Augsburg are multi-sensor datasets (combining Optical, SAR, and LiDAR) characterized by extreme modality heterogeneity and distinct physical imaging mechanisms. Here, standard point-based alignment fails, and our distributional WPCL yields massive gains. XuZhou and Salinas, conversely, are constructed by extracting spatial features (EMP, 3D Gabor) from the same optical sensor. Because these views share an underlying homogeneous spectral source, the initial feature spaces are already structurally similar. Baseline methods can align these relatively homogeneous views effectively, naturally narrowing the performance gap and resulting in the marginal ACC improvements observed.
>
> **Q3:** We wish to explicitly clarify that **we did not inject our privacy-preserving Gaussian perturbation noise into the inputs or prototypes of the baseline methods.**
>
> The baselines were executed precisely as proposed by their original authors, using their native communication protocols, raw features, and inputs that they fully support. The fragmented, visually poor results of the baselines in Figure 3 are not caused by artificial Gaussian perturbation. Rather, what appears as noise in the visual clustering maps are actually classification errors. These artifacts occur because traditional baselines operate on a pixel-by-pixel basis. Our method's superior visual fidelity is not the result of an unfair benchmark designed around noisy prototypes. It is the direct result of our framework explicitly incorporating spatial contextual priors.
>
> **Q4:** We completely agree with the reviewer that placing the raw multi-sensor views adjacent to the clustering results provides a much more intuitive and direct visual comparison. In our final revision, we will update Figure 3 in the main text to integrate the raw remote sensing multi-sensor views directly alongside the Ground Truth and the respective algorithmic clustering maps.
>
> **Q5:** We have included the low-resolution baseline results in the revised appendix, demonstrating that our framework consistently outperforms these methods while uniquely avoiding the destructive spatial information loss inherent to downsampling.
> | Method |  | Trento |  |  |  |  | Salinas |  |  |  |
> |--|--|--|--|--|--|--|--|--|--|--|
> |        | ACC | KAPPA | NMI | ARI | PUR | ACC | KAPPA | NMI | ARI | PUR |
> | FMVC-IMK   | 44.5 | 44.6 | 28.8 | 32.4 | 44.6 | 22.4 | 18.3 | 14.1 | 21.9 | 27.9 |
> | TensorFMVC | 25.1 | 3.8  | 0.8  | 0.3  | 34.8 | 14.4 | 5.9  | 6.2  | 1.8  | 22.1 |
> | MGCD | 78.3 | 71.7 | 72.6 | 71.2 | 86.8 | 72.5 | 70.2 | 71.6 | **64.7** | 72.1 |
> | Ours  | **87.0** | **83.2** | **77.0** | **81.6** | **87.0** | **73.3** | **70.4** | **74.9** | 64.5 | **74.3** |
>
> | Method |  | XuZhou |  |  |  |  | Augsburg |  |  |  |
> |--|-|--|--|--|--|--|--|--|--|--|
> |        | ACC | KAPPA | NMI | ARI | PUR | ACC | KAPPA | NMI | ARI | PUR |
> | FMVC-IMK   | 42.1 | 29.2 | 6.3 | 6.8 | 42.6 | 47.6 | 25.6 | 14.9 | 15.1 | 48.3 |
> | TensorFMVC | 19.7 | 3.5  | 1.4 | 1.0  | 38.3 | 32.2 | 0.2  | 0.1  | 0.1  | 38.7 |
> | MGCD | 54.6 | 47.3 | **52.1** | 40.2 | **69.2** | 35.2 | 21.6 | 18.4 | 10.4 | 54.3 |
> | Ours | **61.0** | **52.3** | 51.5 | **48.7** | 64.8 | **70.1** | **56.7** | **44.1** | **39.8** | **68.9** |

---

> > ### Author Rebuttal · Reviewer_wFy9 · 2026-04-02
> >
> > my main concerns are resolved.

---

> > > ### Author Response · Authors · 2026-04-02
> > >
> > > Thank you for your positive response. We appreciate your time and valuable feedback, which have greatly helped improve our work.
> > >
> > > Best regards,
> > >
> > > Authors

---

### Official Review · Reviewer_rTGs · 2026-03-12

**Soundness:** 4
**Presentation:** 3
**Significance:** 4
**Originality:** 3
**Overall Recommendation:** 5
**Confidence:** 5

**Summary:**

This paper introduces FedRSMVC, a deep federated multi-view clustering framework tailored for remote sensing data characterized by multi-source heterogeneity and strict privacy constraints. The framework mitigates computational overhead via superpixel segmentation and resolves cross-client cluster assignment inconsistencies through a federated co-occurrence structure alignment module. Global semantic consistency is enforced using a Wasserstein prototype contrastive learning mechanism. Data privacy is strictly maintained by exclusively transmitting noise-perturbed prototypes and soft labels, effectively preventing raw feature leakage.

**Compliance With Llm Reviewing Policy:**

Affirmed.

**Key Questions For Authors:**

1. If SAR or LiDAR is chosen as the first view for generating the superpixel segmentation map $M$, instead of optical/high-frequency spectral images, how much will the system's clustering performance and spatial alignment accuracy be affected?

2. In WPCL, negative samples are only selected from other non-corresponding cluster prototypes within the same batch. When the number of clusters $K$ is small, is this weak contrast sufficient to learn a sufficiently discriminative feature space?

3. In Table 2, although FedRSMVC performs well, its runtime is significantly longer than traditional methods such as CeFMC. The authors need to further explain the model's advantage in runtime.

**Limitations:**

Yes

**Strengths And Weaknesses:**

Strengths
1. This work pioneers the application of deep federated multi-view clustering in the remote sensing domain, addressing a critical gap in distributed and privacy-preserving data processing.
2. The integration of superpixel preprocessing drastically reduces the computational complexity of graph construction. Bandwidth consumption is further minimized by communicating compact cluster prototypes instead of high-dimensional gradients or raw features.
3. Empirical evaluations across four public remote sensing datasets demonstrate that the framework surpasses existing federated learning methods and outperforms several centralized methods that require global data access.

Weaknesses
1. The spatial alignment strictly depends on the segmentation map derived from the primary view. A degraded or poor-quality primary view will likely cause cascading spatial alignment failures across all participating clients.
2. The contrastive learning module models clusters using an isotropic covariance matrix $\Sigma = \sigma^2 I$. Since clusters in complex remote sensing feature spaces are frequently non-spherical, this rigid assumption risks significant information loss.
3. Model performance is highly sensitive to the loss balancing weights $\lambda_1$ and $\lambda_2$. Extreme parameter values lead to drastic performance degradation, indicating potentially high hyperparameter tuning costs when deploying the system on datasets of varying scales and complexities.

---

> ### Author Rebuttal · Authors · 2026-03-26
>
> We sincerely thank you for the constructive feedback and insightful questions. We address your specific concerns in detail below.
>
> **W1:** To directly investigate whether our aggregation strategy causes destructive information loss or misalignment in practice, we conducted a rigorous quantitative evaluation of the superpixel segmentation purity across all views. Purity is defined as the percentage of raw pixels within a generated superpixel that share the identical ground-truth semantic label.
>
> The empirical results for all datasets and views are presented below:
>
> | Dataset  | View 1 | View 2 | View 3 |
> |----------|--------|--------|--------|
> | Trento   | 99.52% | 99.68% | -      |
> | Salinas  | 99.84% | 99.81% | 99.00% |
> | XuZhou   | 99.33% | 99.59% | 99.04% |
> | Augsburg | 99.12% | 99.38% | 99.37% |
>
> As the data unequivocally demonstrates, the segmentation purity across all participating clients consistently exceeds 99.00%. This provides a critical validations for our design choice: Spatial Co-registration Overrides Modality Shifts. Because multi-view remote sensing data is strictly co-registered to the same geographical coordinates, the physical boundaries of land-cover objects remain fundamentally consistent across sensors. By designating the modality with the richest spatial-spectral resolution as the primary view to generate $\mathcal{M}$, we capture these definitive physical boundaries.
>
> **W2:** Our adoption of the isotropic assumption, $\Sigma_{i}^{(v)}=(\sigma_{i}^{(v)})^{2}I$, was a deliberate and necessary design choice driven by the strict computational and communication constraints of Federated Learning. Calculating the exact 2-Wasserstein distance between two general Gaussian distributions requires computing the trace term $Tr((\Sigma_{i}^{(v)})^{1/2}\Sigma_{j}^{(u)}(\Sigma_{i}^{(v)})^{1/2})^{1/2}$. For a full covariance matrix, the matrix square root operations incur a prohibitive $\mathcal{O}(d^3)$ computational complexity. The isotropic assumption simplifies this trace to strictly scalar operations, drastically reducing the distance computation to $\mathcal{O}(d)$, which is critical for resource-constrained edge clients.
>
> **W3:** While it is true that extreme parameter values lead to performance degradation, this behavior is theoretically expected. Crucially, the model is not brittle. As demonstrated in our joint sensitivity analysis, the 3D surface plots reveal a broad, stable plateau. Across all four diverse datasets, the clustering performance remains consistently optimal and stable within the broad range of $\lambda_1, \lambda_2 \in [0.1, 1]$. This shared convex-like operational zone empirically proves that FedRSMVC does not require exhaustive, dataset-specific fine-tuning upon deployment.
>
> **R1:** Please refer to the first reply regarding W1 for this section.
>
> **R2:** While instance-level contrastive learning typically requires a massive dictionary of negative samples to prevent mode collapse, our WPCL module operates at the macroscopic prototype level, where the dynamics are fundamentally different.The $K-1$ negative samples in our framework are not weak contrasts; they represent mutually exclusive, high-level semantic clusters. Standard prototype contrastive losses treat centers as deterministic points. In contrast, WPCL models prototypes as isotropic Gaussian distributions $\mathcal{N}(\mu_{i}^{(v)},(\sigma_{i}^{(v)})^{2}I)$ and optimizes the 2-Wasserstein distance. This means the loss actively penalizes the overlap of cluster variances (uncertainties), providing a significantly stronger, geometry-aware gradient signal than point-wise InfoNCE, making a small $K$ highly effective.
>
> **R3:** We acknowledge that traditional, non-deep federated methods like CeFMC possess lower absolute runtimes. However, this speed comes at a severe cost to representation capacity. When benchmarked against other deep architectures, our framework is exceptionally efficient. For example, the deep baseline FedDMVC requires 2565 seconds, while our method converges in just 240 seconds with substantially higher accuracy. The table below shows the runtime of our method compared to federated MVC and centralized MVC, demonstrating that our method has excellent runtime efficiency among deep learning methods.
>
> | Dataset  | FedMVL | FedDMVC | Fed-MVFCM | Fed-MVFPC | CeFMC | Fed-MVKM | MFLVC | MDC | CVCL | TMPCC | Ours |
> |----------|--------|---------|-----------|-----------|-------|----------|-------|-----|------|-------|------|
> | Trento   | 2565.0 | 2565.0  | 34.4      | 1.9       | 3.57  | 32.48    | 949.9 | 573.3 | 654.6 | 3923.5 | 240.19 |
> | Salinas  | 2269.7 | 2255.7  | 182.6     | 24.6      | 19.79 | 55.36    | 1763.5 | 992.5 | 781.0 | 3401.0 | 383.96 |
> | XuZhou   | OOM    | 2886.92 | 365.86    | 28.91     | 11.94 | 87.60    | 1522.4 | 1293.2 | 947.1 | 4100.3 | 323.24 |
> | Augsburg | OOM    | 1588.2  | 438.33    | 27.92     | 9.07  | 222.21   | 1729.2 | 1382.4 | 765.7 | 4837.2 | 374.03 |

---

> > ### Author Rebuttal · Reviewer_rTGs · 2026-04-01
> >
> > My main concerns and problems have been resolved, so I raised my score.

---

> > > ### Author Response · Authors · 2026-04-02
> > >
> > > Thank you for your positive response. We appreciate your time and valuable feedback, which have greatly helped improve our work.
> > >
> > > Best regards,
> > >
> > > Authors

---

### Official Review · Reviewer_uEQN · 2026-03-16

**Soundness:** 2
**Presentation:** 3
**Significance:** 2
**Originality:** 2
**Overall Recommendation:** 3
**Confidence:** 3

**Summary:**

This paper studies clustering of multi-view remote sensing data under federated learning constraints, where data from different modalities (e.g., hyperspectral imagery, SAR, LiDAR) is distributed across institutions that cannot share raw data. The authors argue that centralized multi-view clustering (MVC) is impractical due to privacy restrictions and communication costs, motivating the need for a federated multi-view clustering (FMVC) solution tailored for remote sensing. Experimental results on four remote sensing datasets (Trento, Salinas, XuZhou, Augsburg) show that the proposed method outperforms several federated clustering baselines in terms of clustering accuracy, NMI, and ARI.

**Compliance With Llm Reviewing Policy:**

Affirmed.

**Final Justification:**

Per my comment on the rebuttal, while a number of my concerns are addressed by the rebuttal, I still believe the novelty and theoretical depth of the proposed work is limited.

**Key Questions For Authors:**

It would be great if authors could address the aforementioned concerns raised as weakness. In addition, would the proposed method work on general multi-view datasets (e.g., image-text, multimodal datasets)?

**Limitations:**

Yes.

**Strengths And Weaknesses:**

Strengths
1. Interesting problem setting: The paper tackles a meaningful and relatively underexplored combination of federated learning and multi-view clustering in remote sensing. Remote sensing datasets are often distributed across institutions with privacy constraints, making this setting practically relevant.
2. Empirical performance improvement: The method reports consistent improvements over multiple federated baselines across several datasets and clustering metrics. For example, the model achieves strong gains in ACC and NMI compared to prior methods such as FedDMVC and Fed-MVKM.
3. Ablation studies: The authors include ablation experiments demonstrating the contributions of the two main modules (FCSA and WPCL), which helps clarify their impact on performance.

Weaknesses
1. Novelty of core algorithmic components is limited: Although the paper claims to introduce a new FMVC framework, many core components are adaptations of existing ideas rather than fundamentally new methods: graph autoencoders for clustering are well established; prototype-based federated communication is common in federated clustering and representation learning; contrastive learning with InfoNCE is standard; and Wasserstein distance for distribution alignment has been widely used. The novelty largely lies in combining these components rather than introducing a fundamentally new clustering or federated optimization principle. The claim that this is the “first deep FMVC framework for remote sensing” may also depend strongly on how narrowly the task is defined.

2. Limited theoretical depth: The theoretical justification provided in the paper appears somewhat superficial. For instance, the presented inequality relating the WPCL loss and mutual information is essentially derived from standard InfoNCE bounds and does not provide a new theoretical insight about clustering behavior. Also, the paper does not analyze convergence properties of the federated optimization procedure. Finally, there is no theoretical guarantee that the prototype alignment procedure produces consistent global clusters across heterogeneous clients.

3. Evaluation is limited to a narrow set of datasets: All experiments are conducted on four remote sensing datasets. While these are common benchmarks, they are relatively small and domain-specific. Several concerns arise: no evaluation on general multi-view datasets outside remote sensing; no experiments with varying numbers of clients, which is critical for federated settings; no study of cross-client heterogeneity severity (e.g., non-IID distributions); and no evaluation on large-scale federated systems. This limits the generalizability of the results.

4. Privacy claims are insufficiently justified: The paper claims privacy preservation by adding noise to prototypes before communication. However, no formal differential privacy guarantee is provided. Also, the adaptive noise mechanism does not specify a privacy budget (ε). Finally, no empirical privacy attack evaluation (e.g., membership inference, model inversion) is conducted. Thus, the privacy guarantees appear informal rather than rigorous.

5. Communication efficiency is not thoroughly analyzed: Although the method claims improved communication efficiency, the paper does not present communication cost analysis, bandwidth comparisons, and scaling behavior analysis across clients. Runtime measurements are provided, but they do not directly quantify communication overhead.

---

> ### Author Rebuttal · Authors · 2026-03-31
>
> 1. We respectfully argue that resolving the severe modality heterogeneity in distributed Earth observation data is highly non-trivial and cannot be achieved by naively combining standard components. Our algorithmic novelty lies in two synergistic structural innovations: (1) WPCL: Instead of standard point-based InfoNCE, we model prototypes as Gaussian distributions $\mathcal{N}(\mu_{i}^{(v)},(\sigma_{i}^{(v)})^{2}I)$ and optimize the 2-Wasserstein distance. This uniquely captures and aligns intra-cluster variance across radically different physical sensors. (2) FCSA: We resolve cross-client permutation ambiguity without transmitting raw features by leveraging the isomorphic topological properties of assignment matrices to achieve global index alignment.
>
> 2. On Theorem 1: While building on standard InfoNCE bounds, our key theoretical insight is applying them to the prototype-distribution space. Appendix C proves that minimizing $\mathcal{L}_{wpcl}$ maximizes a tractable variational lower bound of prototype-level mutual information, guaranteeing global separation of heterogeneous clusters. On Convergence: We omitted a formal proof because establishing strict theoretical guarantees under extreme data heterogeneity and partial view alignment would require highly unrealistic assumptions.
>
>
> 3. To verify the generalization ability of our method, we have evaluated FedRSMVC on two widely used general multi-view benchmarks: NUS-WIDE and Scene-15, with the NUS-WIDE dataset consisting of multimodal data including text and images. Our framework consistently outperforms all 9 baseline methods across both general datasets,.
>
> |Dataset|Method|OA|Kappa|NMI|ARI|Purity|
> |-|-|-|-|-|-|-|
> |**NUS-WIDE**|Fed-MVFPC|41.6|24.9|30.8|16.3|43.0|
> ||Fed-MVFCM|39.3|32.5|33.8|16.1|37.4|
> ||FedDMVC|54.0|48.9|39.9|33.2|57.3|
> ||CeFMC|42.5|28.4|29.7|19.7|44.8|
> ||Fed-MVKM|38.4|31.6|23.2|15.5|41.2|
> ||FMVC-IMK|20.7|11.9|6.0|4.3|21.0|
> ||tensorFMVC|11.8|5.0|4.4|2.1|12.2|
> ||FedMVL|42.7|36.4|28.2|21.3|43.8|
> ||MGCD|39.2|32.4|27.7|15.6|42.1|
> ||**Ours**|**60.5**|**56.1**|**47.0**|**41.3**|**61.1**|
> |**Scene-15**|Fed-MVFPC|28.3|19.4|32.0|14.7|29.0|
> ||Fed-MVFCM|37.6|20.4|36.9|20.7|39.8|
> ||FedDMVC|38.6|34.2|36.8|20.3|37.4|
> ||CeFMC|16.3|9.0|12.0|2.3|17.6|
> ||Fed-MVKM|34.6|29.8|35.3|21.1|37.4|
> ||FMVC-IMK|22.8|17.1|22.1|11.7|25.3|
> ||tensorFMVC|10.7|7.2|2.5|1.4|11.2|
> ||FedMVL|32.3|27.4|31.8|15.2|36.6|
> ||MGCD|37.2|32.8|30.4|17.4|39.6|
> ||**Ours**|**41.2**|**37.0**|**38.5**|**22.8**|**44.9**|
>
>
>
> Clarification on the Federated Setup: 0ur framework addresses Vertical-Partitioned FL. In MVC, each client corresponds exclusively to a distinct sensing modality. Because the clients hold strictly co-registered samples of the same physical area but completely different feature spaces, the number of clients is strictly bounded by and equal to the number of available physical views.
>
> 4. We have never claimed that our method has a formal $\epsilon$-DP guarantee. However,  most existing MVC baselines operate with zero privacy protection, transmitting raw data that are highly susceptible to inversion attacks. Compared to these meth, FedRSMVC offers a pragmatic, massively improved security posture via a two-tiered defense.
>
> 5. To directly quantify our efficiency, we have added a comprehensive theoretical and empirical communication breakdown to the revised Appendix. As shown in the table below, the defining advantage of FedRSMVC is that its communication complexity is strictly **$\mathcal{O}(1)$** with respect to the sample size $N$, whereas almost all mainstream baselines scale at $\mathcal{O}(N)$.
>
> | Metric | FedMVL | FedDMVC | FMVC-IMK | TensorFMVC | Fed-MVFCM | Fed-MVFPC | MGCD | CeFMC | Fed-MVKM | Ours |
> | - | - | - | - | - | - | - | - | - | - | - |
> | **Cost** | $12TVNd$ | $8TVN(d+K)$ | $8TVNK$ | $8TVNK$ | $8TVNC$ | $8T\sum(Cd_v)$ | $8TVm(N+d)$ | $8TVK(N+d)$ | $8T\sum(N+K d_v)$ | $\mathbf{4TV (Kd+K^2)}$ |
> | **w.r.t $N$** | $\mathcal{O}(N)$ | $\mathcal{O}(N)$ | $\mathcal{O}(N)$ | $\mathcal{O}(N)$ | $\mathcal{O}(N)$ | $\mathcal{O}(1)$ | $\mathcal{O}(N)$ | $\mathcal{O}(N)$ | $\mathcal{O}(N)$ | $\mathbf{\mathcal{O}(1)}$ |
>
> This $\mathcal{O}(1)$ complexity yields two critical practical advantages:
>
> 1. Taking the massive Augsburg dataset ($N=78,294$) as an example, FedRSMVC mathematically requires only **0.086%** of the communication overhead compared to $\mathcal{O}(N)$ baselines. This proves its exceptional horizontal scalability for large-scale Earth observation archives.
>
> 2. We calculated the minimum bandwidth required to maintain a strict 1-second per-round latency bound ($\tau_{max}=1s$) on our highest-dimensional dataset, Salinas ($K=16$). Baseline methods like CeFMC demand a sustained bandwidth of at least **19.87 MB/s**, risking severe timeout failures in dynamic edge networks. In stark contrast, FedRSMVC requires only **27.0 KB/s**. This demonstrates our framework can operate stably even in highly constrained, hundreds-of-KB/s environments.

---

> > ### Author Rebuttal · Reviewer_uEQN · 2026-04-02
> >
> > I thank the authors for taking the time to carefully address some of my concerns. I still believe the novelty and theoretical depth of the proposed work is limited. That said, given that my other concerns are mostly addressed I am raising my ranking of the paper.

---

### Decision · Program_Chairs · 2026-04-30

**Decision:**

Accept (regular)

**Comment:**

The paper studies federated multi-view clustering for remote sensing data, a setting motivated by practical constraints such as data silos and privacy preservation across distributed institutions. To address this problem, the authors propose a unified framework that incorporates superpixel-based preprocessing for efficient graph construction, a communication-efficient prototype-sharing mechanism, and a structural alignment strategy to mitigate prototype inconsistency across clients.

The reviewers generally provide positive feedback on both the motivation and the technical design. In particular, they highlight the novelty of introducing deep federated multi-view clustering into the remote sensing domain, as well as the coherence of the proposed framework. The use of superpixels for complexity reduction and prototype-based communication for bandwidth efficiency is considered practical and well-justified. Moreover, the combination of co-occurrence-based alignment and Wasserstein geometry is viewed as a thoughtful design to handle cross-client discrepancies.

Although the Reviewer uEQN still gave a weak reject finally, the main concerns are addressed from the authors. Overall, considering the solid motivation, theoretical grounding, and empirical performance, I recommend acceptance of this paper. The authors should incorporate the modifications in the final version.